# Deep Multiple Instance Learning Predicts Gene Expression from Whole Slide Images in Ductal Carcinoma In Situ

**Marek Oerlemans**[1,2] (iD)                    M.OERLEMANS@NKI.NL
[1] *Department of Molecular Pathology, The Netherlands Cancer Institute, Amsterdam, The Netherlands*
[2] *Department of Radiotherapy, The Netherlands Cancer Institute, Amsterdam, The Netherlands*

**Will Harley**[1]                    W.HARLEY@NKI.NL
**Elinor Sawyer**[3]                    ELINOR.SAWYER@KCL.AC.UK
[3] *School of Cancer & Pharmaceutical Sciences, King's College London, United Kingdom*
**Jelle Wesseling**[1,4,5]                    J.WESSELING@NKI.NL
[4] *Department of Pathology, Antoni van Leeuwenhoek, Amsterdam, the Netherlands*
[5] *Department of Pathology, Leiden University Medical Center, Leiden, the Netherlands*

**Esther Lips**[1]                    E.LIPS@NKI.NL
**Joren Brunekreef**[2]                    J.BRUNEKREEF@NKI.NL

## Abstract

Tissue morphology in whole slide images encodes information about underlying biology, which can be read out as gene expression. We use an attention-based deep multiple instance learning model (ABMIL) on foundation model features to predict gene expression from histopathology. We train in a Dutch ductal carcinoma in situ (DCIS) cohort (n=343) and validate on an independent external DCIS cohort (n=184). In the internal cohort, thousands of genes are significantly predicted; performance is reduced on external validation. The predicted expression captures biological signal which is shown through pathway-level analysis. Our results suggest that histopathology encodes partial information about underlying biology.

**Keywords:** Gene expression, histopathology, foundation models, pathways, multiple instance learning

## 1. Introduction

Tissue morphology, as captured in whole slide images (WSIs), reflects the interplay of intracellular biological processes and cell-to-cell interactions. These processes can be quantified through RNA-seq, yielding gene expression profiles. Prior work has demonstrated that deep learning models trained on TCGA data (Weinstein et al., 2013) can predict gene expression from WSIs (Schmauch et al., 2020; Pizurica et al., 2024). We extend this approach to ductal carcinoma in situ (DCIS), a non-obligate precursor to invasive breast cancer. Our contributions are as follows:

- We investigate whether attention-based deep multiple instance learning can predict gene expression from WSIs in a DCIS cohort that is smaller and biologically distinct from publicly available datasets.

- We show that predicted expression partially captures biological signals through Hallmark pathway analysis.

## 2. Methods

We use a dataset of 343 patients from a Dutch cohort with digitized H&E-stained resection slides and RNA-seq from microdissected DCIS tissue, and an external UK cohort of 184 samples (Clements et al., 2022). Gene expression is batch-corrected with ComBat-seq (Zhang et al., 2020) and TMM-normalized (Robinson and Oshlack, 2010) and filtered based on expression level, retaining 16,456 protein-coding genes. WSIs are tiled in tiles of size 224×224 at 0.5 microns per pixel, encoded with UNIv2 (Chen et al., 2024), and 1,000 tiles are randomly sampled per slide. In addition to DCIS cohorts, we also validate on 1106 samples from TCGA-BRCA (Weinstein et al., 2013), as a comparison to literature. We use attention-based multiple instance learning (ABMIL) (Ilse et al., 2018) with mean squared error (MSE) loss on log-transformed expression. We train the ABMIL model from scratch on the Dutch cohort using five-fold cross-validation and report results aggregated across all held-out folds. UK predictions are averaged across all five models.

We evaluate per-gene Pearson correlation between predicted and observed expression, counting genes with significantly positive correlation after Benjamini-Hochberg false discovery rate (FDR) correction. We further assess biological relevance of the predictions by correlating predicted and ground-truth pathway z-scores on the 50 Hallmark pathways (Liberzon et al., 2011).

## 3. Results

We validate our pipeline on TCGA-BRCA and find that we can reproduce the results as presented in literature (Pizurica et al., 2024; Schmauch et al., 2020) using either the proposed methods in the referenced works or a standard ABMIL pipeline.

Table 1 summarizes prediction performance in the Dutch and UK cohort. On the Dutch set, 8,103 genes reach significance at FDR $< 0.05$ with a mean top-1K correlation of 0.27. Performance is substantially lower on the external UK set (80 significant genes at FDR $< 0.05$, mean top-1K correlation 0.21). In the Dutch and UK dataset pathway z-scores from ground truth and predicted gene expressions show a mean correlation of 0.34 and 0.27, respectively.

Table 1: Gene expression prediction validation. Significant genes are FDR-corrected at the shown thresholds. Top-1K is determined based on correlation. Mean correlation for pathways is averaged over 50 Hallmarks.

| Dataset | Significant genes | | Mean gene correlation | | Pathway Z-score |
|---|---|---|---|---|---|
| | FDR $< 0.05$ | FDR $< 0.20$ | Top 1K | All | Mean correlation |
| Dutch (n=343) | 8,103 | 11,296 | 0.27 | 0.10 | 0.34 |
| UK (n=184) | 80 | 1,410 | 0.21 | 0.03 | 0.27 |

Figure 1 (left) shows per-gene correlation distributions. Dots indicate the mean per-gene correlation within selected Hallmark pathways, specifically the five pathways with the highest mean gene-level correlation in the Dutch set. These same pathways also show

above-average correlation in the UK set. The right panel shows scatter plots of predicted versus ground-truth pathway z-scores. In the Dutch set, the top three pathway correlations range from 0.43 to 0.40; in the UK set, from 0.34 to 0.26. These correlations are consistent with the overall pathway mean reported in Table 1.

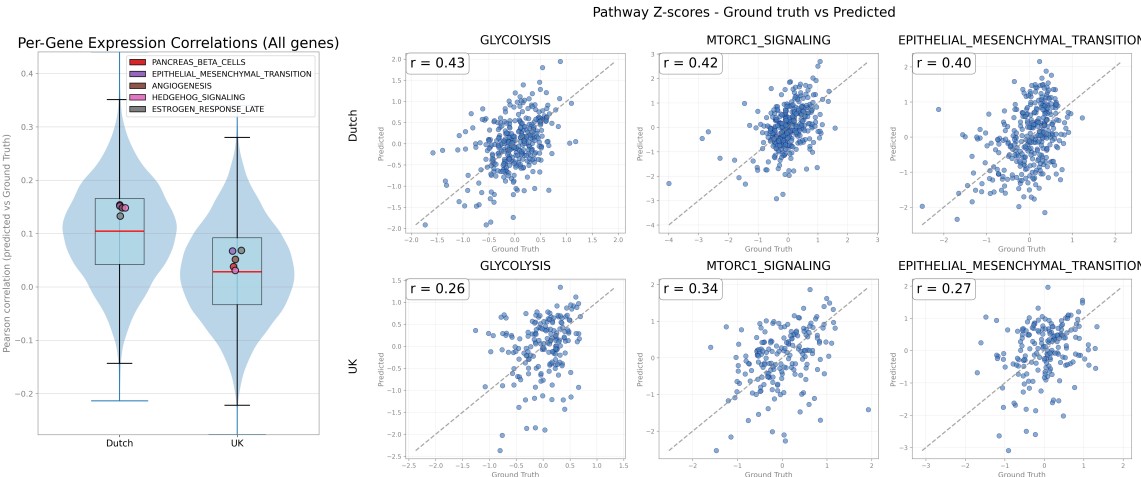

Figure 1: Gene expression prediction validation. Left: per-gene Pearson correlation distributions. Dots indicate the mean correlation across genes within a pathway; the top five pathways ranked by mean gene-level correlation within the pathway on the Dutch set are shown, with the same pathways highlighted on the UK set. Right: per-sample predicted vs. ground-truth Hallmark pathway z-scores. The top three pathways ranked by correlation on Dutch set are shown.

## 4. Discussion and conclusion

We demonstrate that attention-based deep MIL can predict gene expression from WSIs of DCIS patients, we report performance on both internal data and on an independent external cohort. Unlike prior approaches that introduce a task-specific tile-score averaging method (Schmauch et al., 2020; Pizurica et al., 2024), we show the performance of a standard ABMIL approach. We show that predicted genes from DCIS lesions show only modest performance. Generalization to the external cohort is limited. The predicted expressions partially capture biologically meaningful variation, as shown by pathway-level correlations. Pathways whose constituent genes are well-predicted in the Dutch set also tend to be predicted better than average in the UK set, suggesting that the biological signal is consistent across cohorts. Furthermore, pathway scores from predicted gene expression are correlated to those from ground truth gene expression. The trained model can be applied to DCIS cohorts with available WSIs and may offer coarse-grained insight into gene expression and underlying biology. In future work, we aim to relate predicted gene expression to clinical outcomes such as invasive breast cancer recurrence.

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
