# OpenReview forum: "Deep Multiple Instance Learning Predicts Gene Expression from Whole Slide Images in Ductal Carcinoma In Situ"
_MIDL.io/2026/Short_Papers — MIDL 2026 - Short Papers Poster_

### Official Review · Reviewer_W2aN · 2026-05-07
**in situ prediction**

**Rating:** 4
**Confidence:** 5

**Review:**

The paper addresses an important and timely problem in computational pathology: inferring molecular information from routine histopathology images using deep learning. The study is well-motivated, particularly because most prior work on gene expression prediction from WSIs has focused on large public cohorts such as TCGA, whereas this work evaluates the problem in DCIS, a smaller and biologically distinct setting. The paper is technically solid, clinically relevant, and provides useful evidence that transcriptomic information can be partially recovered from DCIS histopathology. The work is likely to be of interest to the computational pathology community.

**Summary:**

This paper investigates whether gene expression can be predicted directly from hematoxylin and eosin (H&E) whole slide images (WSIs) of ductal carcinoma in situ (DCIS) using an attention-based multiple instance learning (ABMIL) framework combined with foundation model features. The authors train and evaluate the approach on a Dutch DCIS cohort (n=343) and validate externally on a UK cohort (n=184), demonstrating that thousands of genes can be significantly predicted internally, although performance drops substantially on external validation.

**Strengths:**

Addresses an important and clinically relevant problem: predicting gene expression directly from histopathology in DCIS, a comparatively underexplored setting.
Includes both internal and external validation cohorts, which provides a more realistic assessment of model generalization than internal-only studies.
Uses modern pathology foundation model features (UNIv2) combined with a simple and interpretable ABMIL framework.
Demonstrates that biologically meaningful information is captured through Hallmark pathway-level analyses, not only per-gene metrics.
Reproduces prior TCGA-BRCA findings, which supports the correctness and robustness of the implementation.

**Weaknesses:**

External generalization performance is substantially weaker than internal validation, raising concerns about robustness and cohort-specific overfitting.
Limited methodological novelty; the work mainly applies an existing ABMIL framework with pretrained embeddings to a new dataset.
Lacks comparisons against stronger contemporary baselines such as transformer-based MIL or domain adaptation methods.
Biological interpretation remains somewhat limited.
No uncertainty estimates or confidence intervals across folds are reported, making result stability harder to assess.

**Justification Of Rating:**

The paper presents a technically sound and clinically relevant study that extends WSI-based gene expression prediction to DCIS using a reasonably rigorous experimental setup, including external validation. The pathway-level analyses add biological relevance, and the manuscript is generally well written. However, the methodological contribution is limited, as the approach largely combines existing ABMIL techniques with pretrained pathology features.

---

### Decision · Program_Chairs · 2026-05-08

Accept (Poster)